# Amyloid-*β* Tetramers and Divalent Cations at the Membrane/Water Interface: Simple Models Support a Functional Role

**DOI:** 10.3390/ijms241612698

**Published:** 2023-08-11

**Authors:** Pawel Krupa, Giovanni La Penna, Mai Suan Li

**Affiliations:** 1Institute of Physics, Polish Academy of Sciences, 02-668 Warsaw, Poland; pkrupa@ifpan.edu.pl (P.K.); masli@ifpan.edu.pl (M.S.L.); 2Institute of Chemistry of Organometallic Compounds, National Research Council, 50019 Sesto Fiorentino, Italy; 3Section of Roma Tor Vergata, National Institute of Nuclear Physics, 00133 Roma, Italy; 4Institute for Computational Science and Technology, Ho Chi Minh City 700000, Vietnam

**Keywords:** ion homeostasis, divalent cations, synaptic plasticity, amyloid peptides

## Abstract

Charge polarization at the membrane interface is a fundamental process in biology. Despite the lower concentration compared to the abundant monovalent ions, the relative abundance of divalent cations (Ca^2+^, Mg^2+^, Zn^2+^, Fe^2+^, Cu^2+^) in particular spaces, such as the neuron synapse, raised many questions on the possible effects of free multivalent ions and of the required protection of membranes by the eventual defects caused by the free forms of the cations. In this work, we first applied a recent realistic model of divalent cations to a well-investigated model of a polar lipid bilayer, di-myristoyl phosphatidyl choline (DMPC). The full atomistic model allows a fairly good description of changes in the hydration of charged and polar groups upon the association of cations to lipid atoms. The lipid-bound configurations were analyzed in detail. In parallel, amyloid-β 1–42 (A$\beta_{42}$) peptides assembled into tetramers were modeled at the surface of the same bilayer. Two of the protein tetramers’ models were loaded with four Cu^2+^ ions, the latter bound as in DMPC-free A$\beta_{42}$ oligomers. The two Cu-bound models differ in the binding topology: one with each Cu ion binding each of the monomers in the tetramer; one with pairs of Cu ions linking two monomers into dimers, forming tetramers as dimers of dimers. The models here described provide hints on the possible role of Cu ions in synaptic plasticity and of A$\beta_{42}$ oligomers in storing the same ions away from lipids. The release of structurally disordered peptides in the synapse can be a mechanism to recover ion homeostasis and lipid membranes from changes in the divalent cation concentration.

## 1. Introduction

Monovalent ions such as Na^+^ and Cl^−^ are at average concentration of approximately 0.14 M in extracellular fluids such as blood plasma [1]. The changes and gradients in the concentration of these ions are fundamental for cell physiology in all organisms. In particular, these ions play a fundamental role in the membrane polarization upon the transmission of signals between neurons [2]. Most of these effects occur in the tiny space of the neuron synapse. Because of their important role, the interactions between monovalent ions and lipid bilayers have been recently revisited by accurate experiments [3,4,5] and models [6,7].

Despite the lower concentration, both in cells and in extracellular space, of the divalent cations compared to monovalent ones, the relative abundance of some divalent cations (Ca^2+^, Mg^2+^, Zn^2+^, Fe^2+^, Cu^2+^) in particular spaces, such as the neuron synapse, raised many questions on the possible effects of free multivalent ions [8] and of the required protection of membranes by eventual defects caused by strong local interactions between divalent cations and lipids [9]. Multivalent cations belonging to the *d*-block and available in cells, such as Fe, Mn, Co, Cu, and Zn, strongly interact with structurally disordered proteins [10,11] that eventually modulate the propensity of moving electrons around the positive holes, such as in reactive events [12]. For instance, copper ions in contact with amyloid-β peptides form catalysts for the production of reactive oxygen species, activating dioxygen molecules [13,14] and promoting oxidative pathways [15,16,17,18]. Though the concentration is very small, divalent cations change the membrane structure and transport properties [19,20,21] and reactivity [22], thus, possibly promoting protein aggregates resembling ion channels and membrane pores [23,24]. Thus, the bouncing of multivalent cations among proteins and lipids is part of the charge transfer and membrane polarization.

Copper and zinc are particularly abundant in the synaptic region [25]. While the physiological Cu(II) concentration within the synaptic cleft and during synaptic vesicle release is 15 μM, it achieves 300 μM concentration upon neuronal depolarization [26,27]. The hypothesis for the copper buffering activity of membrane proteins was proposed for prion (see Ref. [27] and references therein) and amyloid precursor protein (Ref. [28] and references therein). Despite being very low, these concentrations are many orders of magnitude larger than that inside the cell, where Cu, for instance, is present in negligible amounts as an ion available for interactions [29,30]. The discovery that amyloid precursor protein is a copper mediator [31] was lately associated with many neurodegenerative disorders [32,33,34]. As many authors suggested, further studies are required to better understand the molecular pathways that are affected by copper in living neurons [35].

Molecular simulations, particularly molecular dynamics (MD), became a standard tool of computational biology to study molecular interactions in such complex frames [36]. A large number of simulation studies of species involved in neurodegeneration and in contact with membrane models have been reported [37,38,39,40,41,42,43,44,45]. In these studies, the role of cofactors abundant in the environment of neurons were seldom taken into account [46]. Therefore, the role of divalent ions such as copper for a correct physiology of the synapse [34] still requires attention also from a modeling perspective.

Because of these important issues, the modeling of interactions between divalent cations with lipid charged and zwitterionic membranes is becoming a challenge [47,48,49]. Indeed, recent polarizable models explain the experimentally observed strong interactions between Ca^2+^ and phosphate groups in POPC bilayers [49].

The aim of this work is to understand the possible location of free divalent cations that can be released, because of any reason, in the synapse. The membranes limiting the synapse are composed of many different molecules: neutral (such as phosphatidyl choline) and negative phospholipids (such as phosphatydil serine, cholesterol, sphingomyelin, and ganglioside [50] where proteins are embedded. However, the presence of net negative charges, the strong Coulomb interactions between the latter and charged species, and the possible segregation of compounds into patches require extremely large models and sampling. We begun the study by investigating a simple atomistic model containing pure lipid bilayers, water, a background of monovalent cations, and a series of divalent cations of interest in the synaptic space, namely, Mg^2+^, Ca^2+^, Zn^2+^, Fe^2+^, and Cu^2+^. For the first time, we used a recently proposed force-field for such cations [51] in the context of zwitterionic lipids, di-myristoyl phosphatidyl choline (DMPC). These ions are chosen as Mg^2+^, Ca^2+^, Fe^2+^, Cu^2+^, and Zn^2+^. By comparing these ions, copper is found to mimic the binding of calcium. According to this result, we exploited models of amyloid-β(1–42) tetramers to understand the possible role of these protein assemblies in removing cations from the bilayer, with particular attention to copper. In the tetramer assembly, we used dimers as building blocks according to our recent experimental and computational results about oligomerization of Aβ(1–42) in a water solution [52,53]. The systems studied are summarized in Table 1 and Table 2.

The further direction will be to understand the changes in membrane composition that either prevent or enhance dangerous interactions between free divalent cations and lipids, also in cooperation with intrinsically disordered proteins such as amyloid-β peptides.

## 2. Results

### 2.1. Free Energy of Cation Absorption to DMPC (Model M/DMPC)

One configuration of the system representing a single cation of interest, M^*q*+^, in the region of the DMPC bilayer/water interface is displayed in Figure 1 (left panel).

The potential of mean force *F* as a function of the collective variable chosen in the umbrella-sampling (US) simulation, $d_{z}$, is displayed in Figure 2. The $d_{z}$ coordinate of cation M is the difference between the *z* coordinate of M and the average *z* coordinate of P atoms. The *z* axis is parallel to the normal of the bilayer initial plane.

The monovalent cation, Na^+^, is distributed as an ionic atmosphere, since the work to unbind the cation is lower than RT. This is also evident by measuring, in conventional MD (cMD, hereafter), the minimal distance between the cation and atoms in DMPC (see Appendix A). Furthermore, the Fe and Mg divalent cations are weakly bound to DMPC. By inspecting the configurations around the *F* minimum (see also next subsection), the two cations do not lose most of the water molecules in the first-shell hydration sphere. The Ca cation is the one with the position of the *F* minimum below the bilayer average width (the vertical line in the figure). Since the width is measured as the average distance projected along *z* (the bilayer normal), the position of minimal *F* for Ca is deeper than the phosphate groups in the lipid bilayer. Na, Fe, Mg, and Zn display an increase in energy upon desorption from the bilayer that is about RT, while Ca and Cu display a much larger energy change. This shows that Ca and Cu are strongly absorbed over the bilayer surface, while the other cations behave like mobile cations, either as free ions (Na^+^) or hydrated ions (Fe^2+^, Mg^2+^, Zn^2+^). Fe and Zn behave similarly to Mg, showing that all of these ions strongly interact with water molecules.

Ca and Cu have a similar desorption curve, but Ca is able to penetrate deeper into the bilayer. However, the depth of Cu penetration is significant, showing that reactions catalyzed by Cu are potentially more effective than those catalyzed by Fe. Cu is the ion most similar to Ca, and this similarity occurs because Cu is less hydrophilic than all the other ions. The number of water molecules bound to Cu are less than for Mg, Zn, and Fe, mainly because of the force-field that parametrically fits the known coordination number of the ions when in a water solution [51].

### 2.2. Geometry of Divalent Cation Binding Site

The effect of divalent cations is local in most of the cases, and the changes in lipid bilayer observable properties are negligible. We first analyzed the bilayer thickness by the direct measurement of the density of atoms along the direction normal to the bilayer surface (the *z* axis in the simulations) and by using the trajectories collected by cMD simulations. To clarify the nature of the simulated bilayer, we chose different sets of atoms to calculate the density: all atoms except those in water molecules; the addressed cation, that is, the ions different from K^+^ and Cl^−^ that are in the ionic background; all atoms belonging to water molecules. The *z* coordinate is measured with respect to the average *z* coordinate of P atoms, because there is no swap of lipid molecules among the two layers in the bilayer. The values of density *P* are normalized such that $\sum_{i}P_{i}\delta z=1$, with *i* running over the same number of chosen *z* intervals (of size δz), that is, 50 in all cases. The data are then summed up over the equivalent negative and positive *z* values to display only the positive *z* range.

In Figure 3, the atomic density is compared among all simulated systems (with different addressed cations in different colors) and for different atomic sets (different panels). The average location of the P atoms (data not shown for clarity) corresponds to the shoulder of *P* calculated for all non-water atoms (top panel). Therefore, the bilayer thickness is approximately 17 Å for DMPC. Only Cu ions do not move across periodic boundary conditions, showing an absorption propensity higher than Ca. Ca is the cation more deeply penetrating the DMPC bilayer, showing the maximal localization at z∼15 Å thus, even deeper than the localization of P atoms. All divalent cations, differently from Na^+^, are absorbed close to the lipid headgroups. Even though the maximal localization of Cu is at a distance slightly larger than that of the minimal *F* in Figure 2, its density is one half that of Ca when the latter is maximal (∼15 Å).

In Table 3, we report the average order parameters that are measured routinely in NMR after the introduction of the concept of the “molecular electrometer”, proposed years ago on the basis of the α and β choline ${}^{2}$H-NMR order parameters [55]. This measure was proposed as a ruler to assess the computational set-up of cations [56]. We compare the average values measured in MD simulations to the most recent values measured by NMR for a similar system: POPC at *T* = 300 K [57], but in the absence of divalent cations. Though the effect is, in absolute value, very small, we notice that the signs of the order parameters are consistent with the experiments.

Such a local effect of the divalent cation is expected by the low concentration used in this work (approximately 10 mM). This concentration is in that range for Ca and Mg, but it is well above the maximal value expected in the synaptic region for Cu and Zn. In the following, we investigate if, in the cases where cation absorption is strong, the cation shows specific interactions with lipid molecules. We define, hereafter, the absorbed cation as lipid bound and the cation environment in the bound condition as the binding site.

The time evolution in cMD of the minimal distance between each cation and any of the atoms in DMPC (Appendix A) shows that divalent cations exchange between bulk water and DMPC a few times within 1 μs in the case of Mg, Fe, and Zn. Conversely, the binding of the cation to DMPC is rapid, strong, and permanent for Ca and Cu. This limitation of the force-field used for divalent cations is expected in cMD. As for the measure of the free energy of binding to DMPC (Figure 2), this limitation was circumvented by using the combination of steered MD (SMD) and umbrella sampling (see Methods for details).

As for DMPC, the candidate lipid ligand atoms are the oxygen atoms in carbonyl, ester, and phosphodiester groups. These atoms are, in the following, indicated as OC (the carbonyl oxygen in the ester linkage), OCe (the ester oxygen in the same group), OP (the terminal O atom bound to P), and OPe (the O atom in the phosphodiester linkage). In all cases, these ligand atoms compete with the oxygen in water molecules for a correct position into the first-coordination shell of the given cation.

The coordination of these O atoms in the first shell of divalent cations is summarized below, as it is observed in the configurations contributing to the free energy minimum of Figure 2. The representative configurations are displayed in Figure 4 for M = Mg, Zn, Ca, and Cu. The analysis of the radial distribution function, g(r) (Appendix A), obtained by cMD trajectories for pairs involving the addressed cation and O atoms in the DMPC bilayer, shows that the behavior of cations can be divided into three classes. Na is weakly interacting with the bilayer. However, the interaction of Na^+^ with water molecules is also weak, thus, allowing the approach of Na to the oxygen atoms of the phosphate group. In agreement with the rapid exchange of DMPC ligand atoms with bulk water (Appendix A), the Na monovalent cation explores different O atoms in DMPC, spanning a loose layer of approximately 5 Å in size, the latter describing an ionic atmosphere rather than a thin absorbed layer. A similar behavior is shown by K, where the distance between K and OP/OC is only 0.3 Å larger than Na–OP/OC because of the larger size of K compared to Na. As for the radial distribution function, Cu shows a behavior similar to Mg, Fe, and Zn (bottom of Appendix A). However, the US procedure allows for the sampling of configurations where DMPC atoms are in the first-shell coordination sphere of Cu. Mg, Fe, and Zn show at most one DMPC atom in the first shell (top panels of Figure 4 for Mg, left, and Zn, right). The broad M–OP/OC peaks of the radial distribution function at approximately 4 Å (Appendix A) shows that the cations are tightly bound to water molecules that rarely allow the exchange of O atoms in the first-shell coordination sphere with those of DMPC. The configurations (top panels of Figure 4) display five water molecules bound to Mg and Zn and only one OP atom that enters into the first-coordination shell of cations. The second shell peaks for M–OP/OC are visible in g(r) for Mg, Fe, Zn, and Cu, showing that the hydrated cations approach phosphate groups but rarely exchange ligand O atoms belonging to water molecules with those of the DMPC headgroups. A more extended binding of the divalent cation to the O atoms of DMPC is shown by Ca and Cu (bottom panels of Figure 4, Ca, left, and Cu, right). As for Ca, three phosphatidyl groups bind a single Ca cation, and the coordination (six ligand atoms) is completed by three water molecules. In the case of Cu, the first-coordination sphere includes two DMPC molecules, with one OC atom and one OP atom, with four water molecules completing the coordination 5–6 forced by the used empirical potential of Cu^2+^ [51].

The major difference between Mg, Fe, and Zn, on one side, and Ca and Cu, on the other side, is that in the first case, no more than one DMPC headgroup is attracted by the divalent cation, while in the second case, 2–3 molecules are reached. Cu behaves as a smaller probe compared to Ca, the latter rapidly assembling 3 DMPC molecules around the cation, always involving phosphate O atoms. However, the larger propensity to lose water molecules from the first-coordination sphere of Cu than the series Mg, Zn, and Fe is the main reason for the penetration of Cu into the bilayer comparable to Ca.

This behavior shows that Ca and Cu ions both become trapped by DMPC headgroups. At least three water molecules are also drawn towards the headgroups by divalent cations such as Ca and Cu. As observed above, all other cations can rarely, within the 1-μs MD simulation time, exchange the water molecules with atoms belonging to the DMPC headgroups. As for the empirical model used for divalent cations, Cu is the only cation that can easily mimic the behavior of Ca.

### 2.3. Assessing the Geometry of Divalent Cation Binding Site

The empirical model used for divalent cations in cMD has severe limitations due to the low propensity to exchange atoms in the first-shell coordination sphere. Therefore, we used a density-functional theory (DFT) method to assess the coordination of some of the binding sites observed by means of empirical methods. To estimate the change in energy moving each cation from the hydrated sphere towards the possible binding sites involving DMPC headgroups, we measured the change in energy in the following reaction:(1)$$M-{\left(H_{2}O\right)}_{y}^{2+}+xPC^{-}⟶M-PC{\left(H_{2}O\right)}_{\left(y-z\right)}^{(2-x)+}+zH_{2}O$$
where:*y* is the number of water molecules in the bulk hydrated M cation;*x* is the number of PC headgroups assembled around each cation;*z* is the number of water molecules released by the hydrated cation when bonding the assembled PC headgroups.

The potential energy of each component in the reaction above is approximated as the total minimal DFT energy of each state. Since the number of atoms in the system is large, we used a large tolerance for atomic forces in the energy minimization (see Methods).

The energy of all species was calculated in a mean field represented by pure water. The *x* PC^−^ reactant, with total charge q=−x, is assumed already assembled or pre-organized. Therefore, all entropic contributions were neglected, assuming that the organization of both water molecules and PC headgroups around the cation does not change by changing the cation M when the energy change in reaction (Equation 1) is calculated. The entropy changes in reaction (Equation 1) are assumed, therefore, the same is considered for different divalent cations in the studied series, Mg, Ca, Fe, Zn, and Cu. This calculation aims at ranking different candidate binding sites. The results of the energy change in reaction (Equation 1) are in Table 4, for M = Mg, Ca, Zn, and Cu. In the case of Mg^2+^ and Zn^2+^, the interactions with PC headgroups are not sufficient to compete with the hydration of cations. On the other hand, in the case of Ca^2+^, the first-shell water molecules are easily displaced and the cation can penetrate deep into the bilayer, as observed in the free energy profile (Figure 2) described above on the basis of empirical potential energy. Cu^2+^ penetrates into the bilayer and approaches the ester group of lipids while still being in contact with the bulk water at the water/lipid interface. The calculation of the energy change based on the DFT model confirms that Cu behaves similarly to Ca in terms of the replacement of the hydration sphere with the DMPC headgroups. The DMPC-bound configurations corresponding to minimal energy are displayed in Figure 5.

It is interesting to notice that in the case of Cu, the OP–Cu distances increase, with Cu and OP atoms becoming intercalated by Cu-bound water molecules.

The initially Cu-bound OC atom, conversely, is kept at binding distance.

### 2.4. A$\beta_{42}$ Tetramer and DMPC (Models ***1***–***3***/DMPC)

The simulation of the formation of the Cu-A$\beta_{42}$ complex by using the recently proposed divalent-cation model [51] is very challenging, because of the importance of soft interactions such as those between Cu/Fe and His sidechains. Therefore, in this work, we describe the models of the reactant pre-formed A$\beta_{42}$ oligomers (model **1**) and our proposal of the product Cu-A$\beta_{42}$ complex (models **2** and **3**, see also Figure 6).

The representation of the system in one configuration with a tetramer/DMPC interaction to a great extent is displayed in Figure 1, right panel. The behavior of a set of chosen A$\beta_{42}$ tetramers placed close to the lipid/water interface changes according to Cu-loading: thus, we concentrate on this analysis. These changes are described in the following by comparing the results obtained for tetramers in the absence and presence of DMPC. As a reminder: model **1** is a tetramer with no Cu (also indicated with “−Cu”); **2** is a tetramer with one Cu ion bound to a single assembled peptide (“+Cu”); **3** is a tetramer with Cu ions forming bridges between the N-terminus of one peptide and His 13 of the other (“+Cu${}_{b}$”), that is, a tetramer formed by two dimers where covalent bonds are formed between the respective monomers (see Figure 6).

The first studied parameter is the measure of the extent of the contact between the protein assembly and DMPC. We measured the contact by means of the ratio (*R*) between the solvent-accessible surface area of the protein assembly as eventually (R>0) hidden by the lipid bilayer (see Methods). The *R* value was found to be larger than 0.05 in 10.7, 6.8, and 2.6% of the collected configuration for models **1**, **2**, and **3**, respectively. The comparison between the values shows that the Aβ tetramer interacts with the DMPC bilayer more when Cu is not loaded than when Cu is loaded into the protein assembly. A particularly low number of contacts is displayed by the Cu-bridged model **3**. Since the number of trajectories used for these models is lower than that of the models in a water solution, we analyzed the results above, separating the five independent trajectories. In Appendix A, we display the distribution of SASA${}_{\mathrm{PM}}$ and SASA${}_{P}$ for each of the five cMD trajectories. Significant differences can be observed among the different trajectories of each model, as expected with such a low number (5) of samples. On the other hand, the low propensity of protein/DMPC contacts (right panels) when Cu is loaded into the A$\beta_{42}$ tetramers is independent from the trajectory. We notice that some of the **2** and **3** configurations change structure along with time (trajectory 3 in model **2** and 5 in model **3**). In all of these cases the size of the tetramer decreases on a time scale of 0.5 μs. This event occurs because of the low degree of coupling between the protein tetramer and DMPC. Indeed, such tetramer structural changes are hindered when Cu is not bound to the peptide and proteins interact with DMPC (model **1**/DMPC, top panels of Appendix A).

The main effect of DMPC on the peptide composing the tetrameric models is in spreading the configurations of each peptide with respect to that sampled by similar tetramers in water solution. To visualize this effect, we display in Figure 7 the distribution of the minimal root-mean square deviation (RMSD) of each monomer with respect to two different known experimental structures taken as reference. Before comparing the behavior with and without DMPC, we introduce the system in the absence of DMPC (left panels).

The conformation of the C-terminus (region 17–42) in fibrils of Aβ(1–42) is of two types according to fibril structures deposited in the Protein Data-bank (PDB).

The U-like shape of the single-strand fiber (PDB 2BEG [58], solution NMR).The S-like shape of the two-strand fiber (PDB 5KK3 [59] ssNMR, 5OQV [60] cryo EM, 2MXU [61], 2NAO [62]).

As for interactions within each monomer, the major difference between the U- and S-shape peptide is in the formation: in the U-shape, of the Asp 22-Lys 28 intramolecular salt bridge that, in the S-shape, is replaced by the intramolecular Lys 28-Ala 42 salt bridge. The two-strand fiber (S-like shape) displays a head-to-tail approach between the monomers involved in the two facing strands: The N-termini of the two facing monomers are far apart in space, while the N-termini and C-termini are closer. On the other hand, in the single-strand fiber, the side-by-side interactions between monomers involved in the same strand are not diverted by any facing strand. Therefore, the side-by-side interactions between monomers are optimized by approaching the N-termini, even if the latter are structurally disordered. In particular, the Asp 22-Lys 28 salt bridge allows a regular register of hydrogen bonds between backbone groups of the different monomers, structured in parallel β-strands, with N-termini forced to approximately align. In the single strand, N-termini (1–16) are free to move and do not disturb the C-termini that are more involved in the extended β-sheet.

The conformation of peptides in the different S-shape fibers differs mainly in the N-terminus, while the C-terminus is, in all cases, characterized by the sealing effect of the Lys 28-Ala 42 interaction.

Therefore, we used 2BEG and 5KK3 as representative conformations of the U- and S-shapes, respectively, for the peptide C-terminus, region 17–42. In summary, the similarity of a given monomer configuration to the peptide chain in 2BEG means that monomers adopt the shape of those that are involved in side-by-side interactions; the monomer chains similar to monomers in 5KK3 adopt the shape of those that are involved in head-to-tail interactions between the monomers, as in the two facing strands.

In Figure 7, left panels, we display the distribution of RMSD obtained for the four different monomers in the sample (A–D), when the assembly of dimers into tetramers is simulated in the NaCl water solution. Each monomer is compared with the representative monomer in the U-shape (2BEG, top panel for each model) and S-shape (5KK3, bottom of each model). All the configurations (separated dimers and tetramers) are included in the distribution, because no significant difference is found for this parameter when only tetramers are selected. Notably, monomers A and B are involved in pre-formed AB dimers, while C and D are involved in CD pre-formed dimers. This plot allows the following picture for the C-terminus (region 17–42) when the assembly does not interact with DMPC.

The two monomers involved in each dimer behave differently in all cases. Monomers A and C are similar, as well as monomers B and D. A is different from B to the same extent as C with respect to D. When no Cu is bound (“−Cu”, left panels), when monomer A resembles the monomer U-shape, monomer B is more different from the U-shape. The same behavior is displayed by all dimers, also Cu-bound, irrespective of the type of Cu-binding. The low symmetry among A/B and C/D monomers, together with the high symmetry of A/C and B/D, within pre-formed dimers is a result of AB and CD dimer simulation. The similarity between dimers, irrespective of the formation of an assembled tetramer, is the result of the choice of identical dimers as starting points in simulating tetramer formation by random collision between two pre-formed dimers (see Methods).

The deviation of the C-termini from the U-shape is larger when Cu is bound to the peptide, while the binding of each ion to a single peptide decreases the deviation from the S-shape. This effect is linked to the swap of Lys 28 from the Asp 22-Glu 23 negative patch (U-shape) to the C-terminus. The Lys 28-Ala 42 salt bridge can be intramolecular (S-shape) or intermolecular. Therefore, this interaction will be later discussed together with the distance between the N- and C-termini of different monomers.

In the presence of DMPC (Figure 7, right panels, +DMPC), all models display a different behavior, with a distribution of values sparsely spread over a larger range (notice the different *x*-axis range with respect to the left panels) compared to no DMPC. This behavior is characteristic of a strong reduction in symmetry in the macromolecular environment. In particular, the broken C/D symmetry is evident for all models. The monomers embedded in tetramers interact differently with the DMPC/water interface and each monomer A, for instance, starts to change its structure differently from the “twin” (in water solution) monomer C. For instance, with no bound Cu (top-right panel, −Cu), monomer C becomes more similar to 2BEG (U-shape) in region 17–42, while the A monomer maintains a similar deviation compared to no DMPC (lower panel, −DMPC). In the presence of bridging Cu (bottom panels, +Cu${}_{b}$), the deviation from 5KK3 (S-shape) of B and D monomers becomes almost bimodal, with most of the samples at a low deviation (below 7 Å) and other samples at large deviations (13 and 15 Å for D and B, respectively). A similar bimodal distribution is displayed by the tetramer with no Cu for monomers B and D, but with fewer samples at large deviations.

The extent of spreading for the RMSD distribution of monomers when DMPC is added to the sample is an indication of the degree of interaction: when the distribution of RMSD is uni- or bi-modal, the monomer is farther from DMPC, and it is embedded into an environment similar to the water solution. Conversely, when the distribution becomes flatter for a monomer, the monomer is more involved in interactions with DMPC, an environment that is anisotropic and heterogeneous. For instance, with no Cu monomers, A and C interact with the bilayer surface. When Cu is bridging two monomers into dimers, we notice that for all of the monomers, the distribution displays several peaks rather than a broad distribution. Since, in the latter case, the number of contacts with DMPC is smaller, we argue that the smaller the number of contacts with DMPC, the more defined the structure of monomers within the tetramer.

This observation fits with the fact that two covalent bonds connecting different monomers (A–B and C–D) largely constrain the whole monomer structure. On the other hand, since the covalent linkage is between the N-termini (region 1–16), it is interesting to notice that the constraint induced by the linkage is transmitted to the C-termini (region 17–42).

In Figure 8, we display the distribution of some Nt(X)-Ct(Y) distances, with Nt indicating N(Asp 1) of monomer X and Ct C(Ala 42) of monomer Y. This analysis is performed using the configurations of tetramers in a water solution (R≥0.05). We display in different panels the distances within monomers in pre-formed AB and CD dimers (top panels of each model) and between monomers of different dimers (A–C, A–D, B–C, B–D, bottom panels of each model). The presence of DMPC induces a spreading in the AB and CD distributions. In the water solution, monomers within pre-formed dimers are more parallel when Cu is not bound, since both the AB and CD distances display average distances in the range of 2 nm for models **2** and **3**, while the average is approximately 4 nm in model **1**. A small but visible number of Nt(A)-Ct(C) salt-bridges is present in models **1** and **2** but not in model **3**. This small fraction of head-to-tail sealed tetramers is absent only in model **3**, where, conversely, the sealing of pre-formed dimers is enhanced by the presence of DMPC (right-top panel of model +Cu${}_{b}$ in Figure 8). Though a significant approach between Nt(A) and Ct(B) is induced by DMPC when Cu is not bound (top panels of model **1**), the salt-bridge is never sampled. This occurs because the charged N-terminus of A$\beta_{42}$ when Cu is not bound is more available to bind the phosphate groups of DMPC. Therefore, we argue that Cu-binding to A$\beta_{42}$, especially when bridging two different monomers, prevents electrostatic interactions between Aβ-charged groups and the polar lipid headgroups. Despite the DMPC, the lipid is not charged; the net negative charge of protein tetramers drives the Aβ/DMPC interactions.

The number of contacts between the protein and DMPC atoms can be divided among different sets of DMPC atoms and among different residues in the protein assembly. In Figure 9, we display the average number of contacts (one contact when d≤4 Å, with *d* the distance between two selected atoms) for all DMPC atoms (black curve), all OP phosphatidyl atoms (red curve), and O carbonyl atoms in acyl chains (OC, righthand *y*-axis, multiplied by a factor of 1000). It is evident that the tetramer with no bound Cu (top panel) displays a larger number of contacts, especially in the N-terminal regions (r≤16), than the Cu-bound tetramers. The regions of higher number of contacts are more spread over the charged sidechains around the hydrophobic Aβ core (residues 16 and 22–23), when each Cu is bound to each peptide (middle panel). When Cu is bridging two peptides (bottom panel), the number of contacts is the lowest. Even if the contacts involving OC atoms are low in all tetramers (the right *y*-axis is multiplied by 1000), it is evident that OC atoms are more affected by the peptides when Cu is bridging two peptides in covalently bound dimers (bottom panel).

The degree of interaction between the protein tetramer and DMPC is strongly influenced by the orientational degree of freedom of each of the monomers that form the tetrameric assembly. As observed when analyzing the distribution of distances displayed in Figure 8, the partial freedom of monomers to change orientation within the tetrameric assembly when Cu is not bound (model **1**) shows that N-terminus in model **1** is “scratching” DMPC, while in the Cu-bound models, this effect is hindered by the Cu-binding. In the particular case of model **3**, the Cu-bridged model, interactions with DMPC are more localized, resulting in the lowest extent of tetramer hidden surface by DMPC contacts (see above).

Since A$\beta_{42}$ tetramers are characterized by a larger flexibility, due to the absence of the Cu-binding constraints, we analyzed in more detail if important groups of atoms are forced, by the presence of DMPC, to be closer in space. In particular, we analyzed the probability of sampling distances between His sidechains and between salt-bridges that are important in sealing fibril structures (see discussion above). Again, we compare the probability measured when DMPC is present (+DMPC) to the same system when DMPC is absent (water solution, −DMPC). In Figure 10, we display the difference between the number of particles at distance *r* between pairs of atoms in selected groups, where the difference is between the measure for +DMPC and −DMPC (always model **1**). In the top panel, the two identical groups are composed of all Nδ1 and Nϵ2 atoms of His sidechains. In the bottom panel, one group is composed of N(Asp 1), Nζ(Lys 28) (positively charged); the second group is composed of Cδ(Glu 22), Cγ(Asp 23), and C (Ala 42) (negatively charged). All possible distances are counted, meaning that CN is one when one atom in the first group sees one atom of the second group at the given distance *r*. For instance, Nδ1(His 6) certainly shows Nϵ2(His 6) at distance 2.8 Å, being two atoms part of the same His sidechain. We notice that both groups of atoms are made, on average, closer in space when DMPC is present. As for His sidechains, the higher chance to have imidazole N atoms close in space increases the chance to capture metal ions, especially Cu^2+^, which binds at least two His sidechains when absorbed by A$\beta_{42}$ peptides. The selected salt-bridges analyzed in the bottom panel of Figure 10 indicate that those interactions stabilizing U- and S-shapes are better sampled in the presence of DMPC. These observations, taken together, support that the A$\beta_{42}$ peptides, once assembled into tetramers and once close to the DMPC bilayer, become more suitable to bind Cu ions and to fulfill those salt-bridges that seal fibrils, either single (U-shape) or double stranded (S-shape).

## 3. Discussion

Understanding the possible effects of free divalent cations, in particular Cu^2+^ and Zn^2+^, on lipid membranes is of great interest because of the unusually high concentration in the neuron synapses of free divalent cations. Despite the space being crowded by ion-binding proteins, the control of free cations must be constrained to avoid any possible damage caused by those cations that can penetrate into the bilayer and/or catalyze oxidation of lipids.

The development of accurate models for divalent cations in computer simulations is a challenging arena (see Introduction). We used a recent force-field [51], finding, as expected, its severe limitations due to the slow exchange of ligand atoms in the first-shell coordination of the cations. We notice that from ${}^{17}$O-NMR experiments [63], the time spent by water molecules in the first-shell coordination sphere of cations is of the order of μs for Mg^2+^ and Fe^2+^, dropping to ns for Cu^2+^. Therefore, it is not surprising that in a μs-long conventional MD of hydrated cations in contact with a potential binding site, one observes a slow removal of water molecules from the first-shell coordination for ions such as Mg^2+^. On the contrary, for Cu^2+^, which, experimentally, has a short residence time of water molecules in the first-shell coordination sphere, the potential binding site in DMPC allows for the extraction of the cation from the water hydration sphere, but only if a proper statistical method is used. The umbrella sampling method we used to estimate the M/DMPC binding free energy partially overtakes the limitations of the force-field. Furthermore, the DFT calculations provided an alternative view of the same property. Therefore, we conclude that it is important to use many methods to partially compensate for the statistical limitations of each method, in particular, for conventional MD.

As for divalent cations, including copper and zinc, there are a few experimental in vitro studies addressing direct M^*q*+^/DMPC interactions.

The former X-ray studies of Cu^2+^ in contact with DMPC [20] showed that the effect at a low concentration is limited to polar headgroups in a water/DMPC interface.

Even if the effect in DMPC is limited to this region, electrophysiology clearly showed that Cu^2+^ drastically changes the Na^+^ permeability of cell membranes. It is interesting to notice that in the presence of amyloid-β peptides, the alteration of the water/DMPC interface is reduced, while for peculiar cations such as Al^3+^, the formation of hydrophobic complexes that penetrate into DMPC becomes evident [64].

Recent small-angle neutron scattering (SANS) experiments showed that in the presence of a small concentration of divalent cations, the thickness of the DMPC bilayer is significantly increased [65]. However, a deeper analysis of the reported data shows that the main difference between Na and the other cations reported (Mg, Ca, Co^2+^) is at low *q* values. This difference was interpreted on the basis of the screened interaction between spherical vesicles when the divalent cations are added to the solution containing monovalent cations. Indeed, the latter observation indicates that when divalent cations are strongly absorbed, vesicles collapse more easily. Since the sample we simulated is small, we can not even estimate the propensity for stronger interactions between different bilayers, but this is definitely a direction for further modeling studies: our small models indicate that the major effect at a low concentration is to neutralize charge repulsion between bilayers by strong absorption. To confirm this, a more reliable bilayer with negatively charged lipids, such as phosphatidylserine, must be modeled.

In our previous model describing A$\beta_{42}$ and Cu-A$\beta_{42}$ in the monomeric form in contact with a DMPC bilayer [9], we discussed the literature reports about the effect of divalent cations on DMPC. By solid state NMR experiments (ss-NMR), the effect of the addition of free Cu^2+^ and Zn^2+^ ions on the membrane properties was found stronger than in the presence of the Aβ peptide [66]. The effect was monitored by the ${}^{2}$H and ${}^{31}$P spectral perturbation upon the addition of either divalent cations or A$\beta_{42}$. Similar strong effects of free divalent cations have been observed both experimentally and computationally for free Ca^2+^ ions [47,48,49], and Mg and Cu divalent cations are even smaller than Ca in size.

We can summarize some known facts as follows.

Free divalent cations perturb polar and charged bilayers mainly by strong charge neutralization [20,65,66].The same bilayers are more affected by amyloid fibrils than by amyloid peptides in soluble forms [67,68].Amyloid oligomers start perturbing lipid bilayers in computational models, including realistic membrane models, when dodecamers [50].

Even though the effects of oligomers depend on their concentration and composition, the composition of the bilayer, and the presence of cofactors, including divalent cations, it seems that in soluble forms consistent with recent experiments [53], A$\beta_{42}$/lipid interactions are limited to the lipid/water interface. Since the peptides released in the synapse by APP hydrolysis are negatively charged intrinsically disordered (IDPs) and such charged IDPs are known to be particularly avid in ions such Ca^2+^ [69], it is expected that Aβ oligomers can exert a buffering of ions that can perturb, when in a free form, the lipid-charged headgroups.

## 4. Methods

Three types of systems were subjected to the all-atom molecular dynamics (MD) simulations:Cations situated near the DMPC lipid bilayer (models M/DMPC);Pairs of docked A$\beta_{42}$ dimers merged in water, simulated both with and without the presence of copper ions (models **1**–**3**);Tetrameric A$\beta_{42}$ structures, also with and without copper ions, positioned in close proximity to a DMPC (1,2-dimyristoyl-sn-glycero-3-phosphocholine) lipid bilayer (models **1**–**3**/DMPC).

### 4.1. Divalent Cations and DMPC (Models M/DMPC)

The first study consisted of 1 μs conventional MD (cMD) simulation to study the behavior of each of the ions at equilibrium. A steered MD (SMD) was then performed to obtain the initial configurations for the umbrella sampling (US). The latter method is used to obtain the potential of mean force (PMF) as a function of ion distance from the center of the lipid bilayer, projected along the *z* axis. The different systems are summarized in Table 1, and the simulation methods are summarized in Table 2. The details of the simulations are described in the following.

### 4.2. Molecular Dynamics Parameters

All described MD simulations were performed with the Amber20 package [70]. The analysis was performed with AmberTools, VMD [54], Gromacs [71,72], and other tools developed by us. The force-field was FF14SB [73] for peptides, combined with the TIP3P water model [74] and LIPID14 for lipid molecules [75]. Only MD simulations of ions and DMPC were performed with the new force-field for ions, including divalent cations in the 12-6-4 form [51]. The binding of copper to A$\beta_{42}$ was described as in our previous works on Cu-A$\beta_{42}$ monomers, dimers, and tetramers [9,14,76,77,78]. The FF14SB force-field was selected as it is a reliable set parameters to study monomeric forms of A$\beta_{42}$ [79], as well as the amyloid peptide assembly [80].

The analytical concentration of divalent cations we used in simulations is ∼ 0.01 M, while that for the monovalent cations (KCl) is approximately 10 times larger. Therefore, [K^+^] was in the range of the usual experiments, while [M^2+^] was larger than that of interest in cell physiology. However, the conditions used for divalent cations correspond to an isolated cation.

The potential of mean force (PMF) was computed by using, first, SMD, followed by the application of the US method [81]. SMD was begun by placing the addressed cation at 35 Å along *z* from the center of the equilibrated DMPC bilayer. This distance, $d_{z}$, was used as a collective variable in SMD, moving the distance at a pulling velocity of 1 m/s. During this process, in order to avoid the drift of water molecules into the bilayer, we used a restraining harmonic force with a constant of 10 kcal/mol/Å${}^{2}$ applied to all heavy atoms. This SMD trajectory produced the initial configurations for the following US.

The $d_{z}$ collective variable was divided into 70 windows in the range between zero and 35 Å, with a step in $d_{z}$ of 0.5 Å. The initial configurations with $d_{z}$ close to the center of each window were extracted by the SMD simulation. The system was finally simulated in each $d_{z}$ window by using a harmonic potential $U=\frac{k}{2}{\left(d_{z}-d_{z,eq}\right)}^{2}$, with k=20 kcal/mol/Å${}^{2}$ and an equilibrium distance $d_{z,eq}$ at the center of each window. No position restraints were used in US. The center of the bilayer was always computed as the center of the P atoms in DMPC. The lipid molecules (32 per layer) never changed layers during MD simulations. In each $d_{z}$ window of the bias potential, 10 ns of MD were performed, neglecting the first 100 ps as equilibration. Most of the MD parameters are described below.

The conventional MD (cMD) simulations in the NVT ensemble were used to sample configurations close to the free energy minimum determined by the PMF analysis. The temperature was set to *T* = 310 K for a simulation time of 1 μs in NVT after equilibration in the NPT ensemble for 1 ns.

The conventional MD simulations of peptides, both in the water solution and close to the DMPC bilayer, were performed with the following parameters. The temperature was set to 310 K with a Langevin thermostat with a collision frequency of 2 ps${}^{-1}$. The pressure was set to 1 bar with the usual barostat. The system in the water solution was kept at a constant pressure by isotropic volume scaling, keeping the simulation cell cubic with a relaxation time of 1 ps. The simulation cell of the peptide/DMPC system was kept orthogonal with anisotropic cell scaling. All bonds involving hydrogen atoms were kept rigid by using constrained dynamics (SHAKE), allowing a time-step, in all simulations, of 2 fs. The particle-mesh Ewald method was used for Coulomb interactions, using a 9 Å distance cut-off in real space ($R_{c}$), which is the same cut-off used for Lennard–Jones interactions.

The simulation length in the NPT statistical ensemble for peptides in water solution was 120 ns after 2 ns of equilibration. The cubic cell kept a side of approximately 100 Å. We used 128 independent simulations where the dimers were initially separated and randomly oriented. A few of the initial configurations failed to achieve a stable trajectory because of atomic clashes and, therefore, were rejected from the statistics. The number of independent trajectories effectively used in the final statistics is displayed in Table 2, together with a summary of all systems investigated in this work. The cumulative simulation time for tetrameric A$\beta_{42}$ in water was equal to 15.36 μs, with a total of 46.08 μs across all variants.

As for peptides close to the DMPC bilayer, the area per lipid was kept around an average value of 66 Å${}^{2}$, consistently with the DMPC bilayer at *T* = 310 K and *P* = 1 bar at a lipid concentration typical of large unilamellar vesicles. The average cell side along the *z*-axis was approximately 115 Å, thus, containing the bilayer (with a size of approximately 35 Å) and a large layer of water molecules and ions. The maximal gyration radius $R_{g}$ of A$\beta_{42}$ tetramers in the water solution was approximately 18 Å. Therefore, the peptide/DMPC system can contain tetramers sufficiently far ($d>R_{g}+R_{c}$) from the lipid bilayer to be unaffected by real-space interactions.

Since the A$\beta_{42}$/DMPC system is larger in size than the system in the water solution, a necessarily smaller number of independent trajectories was used.

Five initial configurations were used for each model, wherein the tetramer was positioned at a random orientation and at a significant non-interacting distance from the DMPC bilayer. Each individual simulation was equal to 2.5 μs, resulting in a cumulative time of 12.5 μs per system and a total simulation time of 37.5 μs across all variants. The configurations were captured at intervals: every 10 ps in the water solution and every 20 ps in the presence of DMPC. For statistical analysis, only the second half of the trajectories from the Aβ/DMPC systems were used. The distributions of SASA and the related parameters (see below) were compared among each of the different trajectories (five simulated) in Appendix A.

### 4.3. Refining DMPC Ion-Binding Sites

A limited number of representative configurations, close to the free energy minimum identified by the used force-field, were chosen for more detailed calculations, to understand cation coordination in lipid molecules. The configurations were chosen with the distance $d_{z}$ from the center of the bilayer (the same variable used in umbrella-sampling), corresponding to the free energy minimum and with the minimal external bias in the umbrella-sampling.

We used the parallel version of the Quantum-Espresso package [82] to minimize the atomic forces in the geometries obtained with the MD approximations (see above). The DFT approximation is here designed to speed up the calculations, therefore, it involved the Vanderbilt ultra-soft pseudopotentials [83] and the PBE exchange-correlation functional [84]. The electronic wave functions were expanded in plane waves up to an energy cutoff of 25 Ry, while a 300 Ry cutoff was used for the expansion of the augmented charge density in the proximity of the atoms, as required in the ultra-soft pseudopotential scheme. All of the calculations were performed with the contribution of plane-waves with *K* = 0 in the super-cell lattice described by the periodic boundary conditions used, i.e., in the so-called Γ-point approximation of solid-state electron density. We inserted each configuration in a cubic super-cell of 25 Å, suitable to minimize interactions between periodic images of the model complexes. A variable numbers of steps of energy minimization, performed with the Broyden–Fletcher–Goldfarb–Shanno algorithm, were required to achieve atomic forces with any component within 0.01 Ry/bohr. The maximal number of required steps was 120.

The energy values used to compute the energy change in reaction Equation (Equation 1) (see below) were calculated using the implicit solvation scheme implemented in the Quantum-Espresso code [85]. The environment was described as bulk water at room conditions, being the system close to the DMPC/water interface. The energy tolerance for energy change was, as in structural relaxation, 10${}^{-6}$ Ry.

### 4.4. A$\beta_{42}$ Tetramers in a Water Solution (Models ***1***–***3***)

We describe the following in more detail, the construction of models, because this part is important to limit the huge sampling required when intrinsically disordered peptides are involved.

The construction of tetramers started from the simulation of dimers. We worked under the assumption that the building block of A$\beta_{42}$ oligomers are dimers rather than monomers. This was indicated by the observation that in the background of oligomeric assemblies with morphologies that are almost identical when Cu ions are absent or present at 1:1 Cu:A$\beta_{42}$ stoichiometry, the distance between Cu centers measured by ESR is consistent with a particular coordination of Cu to A$\beta_{42}$ [53]: Cu ions are bound to the N-terminus of one peptide and to one of the His sidechains in the 13–14 segment of another peptide, with Cu forming a covalent bridge between two peptides. We indicate this dimer as the Cu-bridged dimer, [Cu-A$\beta_{42}$]${}_{2}$. The fact that this Cu-Aβ dimer is observed by ESR at aggregating conditions, the morphology of aggregated particles and the toxicity of oligomers, all observed also in the absence of Cu ions, indicate that the background of amorphous particles made of soluble oligomers is likely made of dimers with the topology of the Cu-bridged dimers: The N-termini of different peptides are entangled so as to divert C-termini by forming stable pre-fibrillar oligomers.

According to this hypothesis, we first simulate the formation of dimers in a water solution. Then, we combine dimers in samples where the formation of tetramers is allowed. The most abundant tetramers formed in these conditions are then used as initial configurations placed in the water layer close to the lipid DMPC bilayer constructs. Three different models of A$\beta_{42}$ dimers are used:2 × A$\beta_{42}$ (model **1**, hereafter);2 × Cu-A$\beta_{42}$ (model **2**);[Cu-A$\beta_{42}$]${}_{2}$ (model **3**);

The three models are summarized in Figure 6.

The initial configurations of dimers were extracted by previous simulations performed by us [76,77]. As a reminder, for the Cu-coordination, both models **2** and **3** are consistent with the ESR spectrum of the most abundant species at conditions close to physiological ones.

Two pairs of dimers for each model were then inserted into simulation cells, as described above. A number of 128 initial configurations of pairs of dimers were prepared for each of the three models to perform the same number of independent MD trajectories.

In summary, these simulations provide a large number of possible collisions between pre-formed dimers to form tetramers. The eventual formation of tetramers was monitored as in the following.

### 4.5. Dimer/Dimer Contact

We measured the solvent-accessible surface area (SASA) [86] of the protein assembly. A probe of 1.4 Å was used, representing an approximately spherical water molecule. This SASA is indicated as SASA${}_{\mathrm{ABCD}}$, with ABCD indicating the assembly formed by AB and CD dimers, used as the starting configurations in the assembly of tetramers. The extent of the hidden surface when the tetramer is formed is measured relative to the SASA of each bare protein dimer, SASA${}_{\mathrm{AB}}$ and SASA${}_{\mathrm{CD}}$:(2)$$R=\frac{{\mathrm{SASA}}_{\mathrm{ABCD}}}{{\mathrm{SASA}}_{\mathrm{AB}}+{\mathrm{SASA}}_{\mathrm{CD}}}\;.$$

The ratio *R* is one when the dimers is separated. When this ratio *R* is smaller than 0.98, we assign the configuration to a protein tetramer.

### 4.6. A$\beta_{42}$ Tetramers and DMPC Bilayer (Models ***1***–***3***/DMPC)

A set of five initial configurations of A$\beta_{42}$ tetramers, both without and with Cu, was obtained by analyzing the statistics of the respective tetramers in a water solution. The representative five tetrameric models were obtained by the clustering of the subset of the tetramers with the *R* parameter (see previous subsection) lower than 0.9. The first five most populated clusters were selected and each of the five trajectories started from the representative structure of each of the five clusters, respectively. Each configuration of tetramer was placed into the water layer facing DMPC with no heavy atom closer than 10 Å from any heavy atom in DMPC.

Because of the simulation algorithm, which is conventional MD, peptides do not dissociate from the pre-formed tetramers, even though, in theory, they are allowed to do. Therefore, by this method, we can monitor the extent of the interaction between different tetrameric configurations with the DMPC bilayer. The formation of contacts was measured as in the following.

### 4.7. Protein/Lipid Contact

We measured the solvent-accessible surface area (SASA) of the protein assembly when it is placed close to the lipid bilayer. This quantity is indicated as SASA${}_{\mathrm{PM}}$, the latter measuring the protein surface that can be covered by water (P) once part of it is hindered from water by the presence of lipid molecules belonging to the bilayer (M). The extent of the hidden surface is measured relative to the SASA of the totally solvated protein, SASA${}_{P}$:(3)$$R=\frac{{\mathrm{SASA}}_{P}-{\mathrm{SASA}}_{\mathrm{PM}}}{{\mathrm{SASA}}_{P}}\;.$$

The ratio *R* is zero when no contacts are formed between the protein assembly and the lipid bilayer. When this ratio *R* is larger than 0.05, we assign the configuration to a protein/lipid (PM) interacting state.

All computed distributions *P* are normalized as $\sum_{i}P_{i}=1$, with the exception of density profiles.

## 5. Conclusions

In this work, we extended the previous models, the application of which suggested we propose a protective role of the A$\beta_{42}$ monomer in the synapse. We, first, extended the description of interactions between a polar lipid bilayer model, di-myristoyl phosphatidyl choline (DMPC), and divalent cations. We used the most recent model of cations [51], and for the first time, we applied the latter model to DMPC. We also refined selected configurations with computational models including electrons at the usual density-functional level of approximation. These models show that Cu^2+^ cations behave similarly to Ca^2+^ in terms of the type of interactions with DMPC and the strength of binding. On the other hand, Mg^2+^, Fe^2+^, and Zn^2+^ more strongly keep their respective hydration spheres.

We then extended our construction of A$\beta_{42}$ oligomers, extracting realistic models of tetramers, also including bound copper ions in different topologies consistent with the ESR data. We found that A$\beta_{42}$ tetramers with no Cu loaded interact more efficiently with the DMPC headgroups, showing the possibility to reach possible ion binding regions and to extract divalent cations from the lipid/water interface. On the other hand, we found that when A$\beta_{42}$ tetramers are loaded with Cu, the propensity to scratch the lipid/water interface becomes smaller. Most of the effects can be interpreted in terms of the different availability of charged groups to form mutual electrostatic interactions.

These models provide further support to the following hypothesis: the formation of the M^*q*+^-Aβ42 complex, before the increase in peptide concentration, whatever the reason, and before the eventual incorporation into the membrane of large aggregated forms, appears as a protection against membrane destabilization and oxidation by free divalent cations. The latter events can be frequent in the synapse because of the high concentration of free forms of copper, zinc, and iron ions.

## Figures and Tables

**Figure 1 ijms-24-12698-f001:**
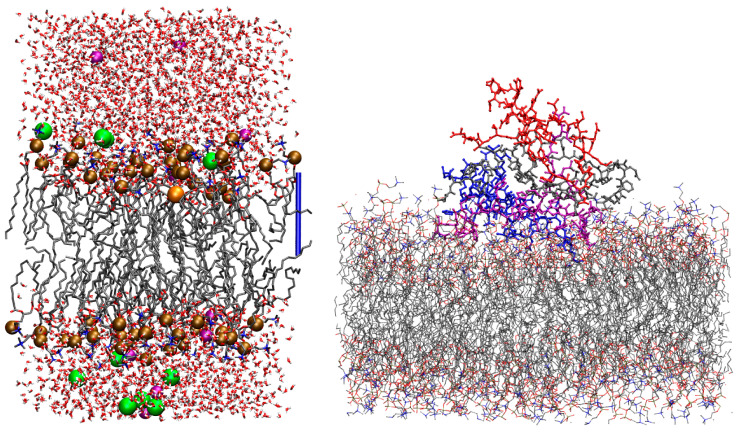
(**Left**) one configuration of the Na^+^/DMPC system obtained by MD, where the Na^+^ cation is deep in the bilayer interface. The ions of interest are represented as spheres, while other components are represented as sticks. C atoms are gray, N blue, P bronze, O red, H white, K purple, and Cl green. The hydrogen atoms of water molecules are displayed, while those in DMPC are omitted for clarity. The addressed cation is the orange sphere. The atomic and bond radii are arbitrary. The bar on the right-hand side shows the *z* axis as a ruler of length 17 Å. VMD [54] is used for molecular drawings. (**Right**) one configuration of A$\beta_{42}$ tetramer displaying a large interaction with DMPC (*R* = 0.1), model **1** (no Cu loaded). A$\beta_{42}$ monomers are A (red), B (purple), C (blue), and D (gray). Hydrogen atoms, water molecules, and ions are not displayed.

**Figure 2 ijms-24-12698-f002:**
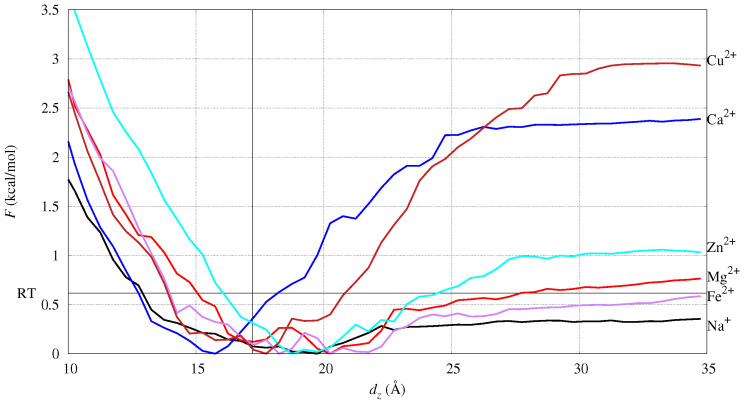
The potential of the mean force (*F*) as a function of the collective coordinate $d_{z}$ used in umbrella sampling. The $d_{z}$ coordinate of cation M is the difference between the *z* coordinate of M and the average *z* coordinate of P atoms. The horizontal black line is F=RT. The vertical black line is at $d_{z}$ = 17 Å, that is, 1/2 the average distance, along with *z* between the P atoms of the two different layers. The zero of *F* is the minimum of *F* observed for each cation.

**Figure 3 ijms-24-12698-f003:**
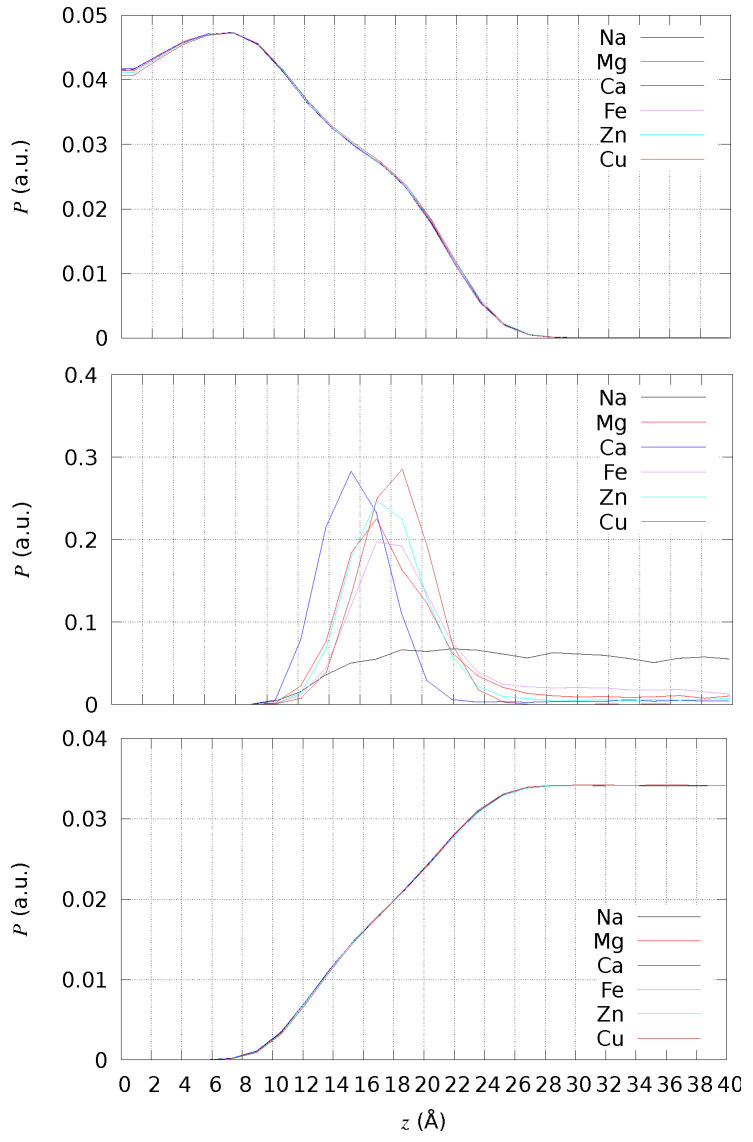
Number density of solute atoms (**top**), addressed cation (**middle**), and water atoms (**bottom**) in DMPC bilayers with different addressed cations. As for comparison, *P* is normalized to have $\sum_{i}P_{i}\delta z=1$, with *i* running over the number of intervals in *z*, and the number of intervals in *z* is the same (50) for different cations.

**Figure 4 ijms-24-12698-f004:**
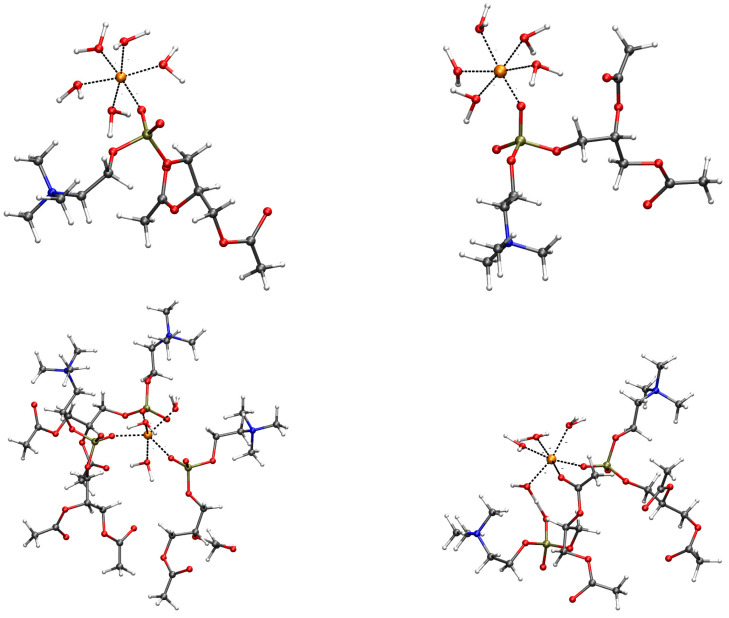
Configurations contributing to the minimum of free energy displayed in Figure 2. Only atoms and residues belonging to the first-coordination sphere of the addressed cation are displayed. The long acyl chains of DMPC are replaced with acetyl groups, as in the following DFT geometry relaxation (see Figure 5). The distances of the first-shell coordination are drawn as black dashed lines. The addressed cations are orange spheres: Mg (**top-left**); Zn (**top-right**); Ca (**bottom-left**); Cu (**bottom-right**). C is gray; O is red; N is blue; P is bronze; H is white. The atomic and bond radii are arbitrary.

**Figure 5 ijms-24-12698-f005:**
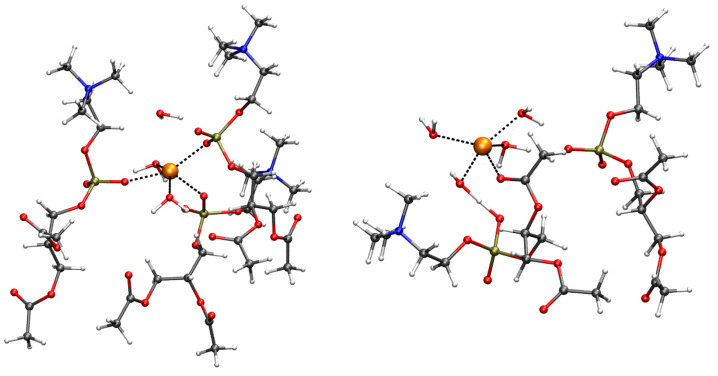
The minimal energy configurations of Ca (**left**) and Cu (**right**) binding sites, obtained starting from the configurations contributing to the minimum of free energy displayed in Figure 2 (Figure 4). Three different phosphatidyl choline (PC) headgroups are assembled around Ca (**left**). Cu is coordinated with OC and four water molecules, with two phosphatidyl choline headgroups assembled via intercalated Cu-bound water molecules. The long acyl chains of DMPC are replaced with acetyl groups. The distances in the first-coordination sphere of the addressed cation are drawn as black dashed lines. Ca (**left**) and Cu (**right**) are orange spheres; C is gray; O is red; N is blue; P is bronze; H is white. The atomic and bond radii are arbitrary.

**Figure 6 ijms-24-12698-f006:**
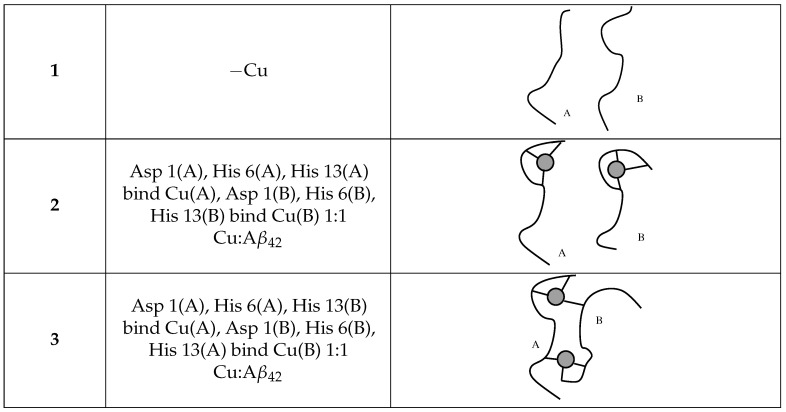
The scheme of Cu-binding for different A$\beta_{42}$ (represented as a curve) dimers, with indices (A, B) used through the text. Cu is displayed as a gray circle.

**Figure 7 ijms-24-12698-f007:**
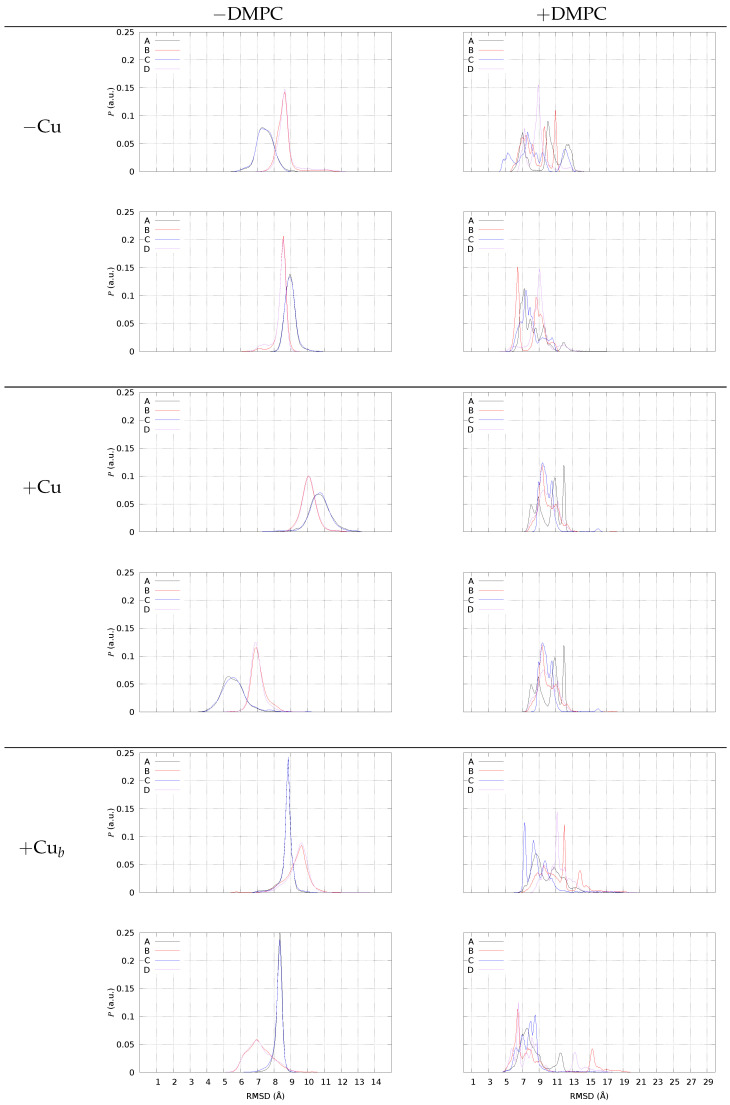
The distribution of minimal root-mean square deviation (RMSD, see Methods) using backbone atoms N, Cα, C, O in region 17–42 of 2BEG PDB [58] (top panels in each model’s frame), and 5KK3 PDB [59] (bottom panels in each model’s frame).

**Figure 8 ijms-24-12698-f008:**
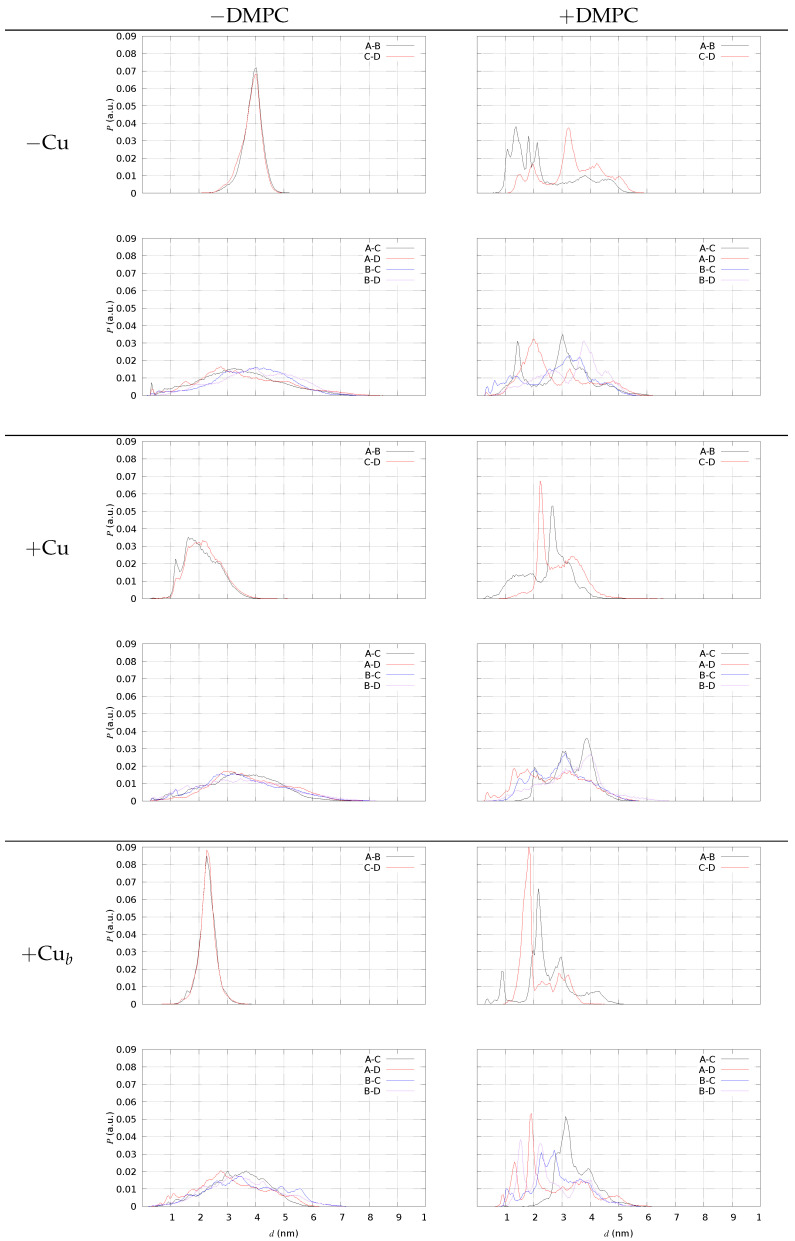
The distribution *P* of distance *d* between N(Asp 1) and C (Ala 42) belonging to different pairs of monomers (A–B, …). Left panels—tetramers (R≤ 0.98) in the water solution; right panels—tetramers (all configurations) with DMPC. Model **1** (−Cu); model **2** (+Cu); model **3** (+Cu${}_{b}$). Top in each frame—distances within pre-formed dimers; bottom in each frame—distances between monomers in different pre-formed dimers.

**Figure 9 ijms-24-12698-f009:**
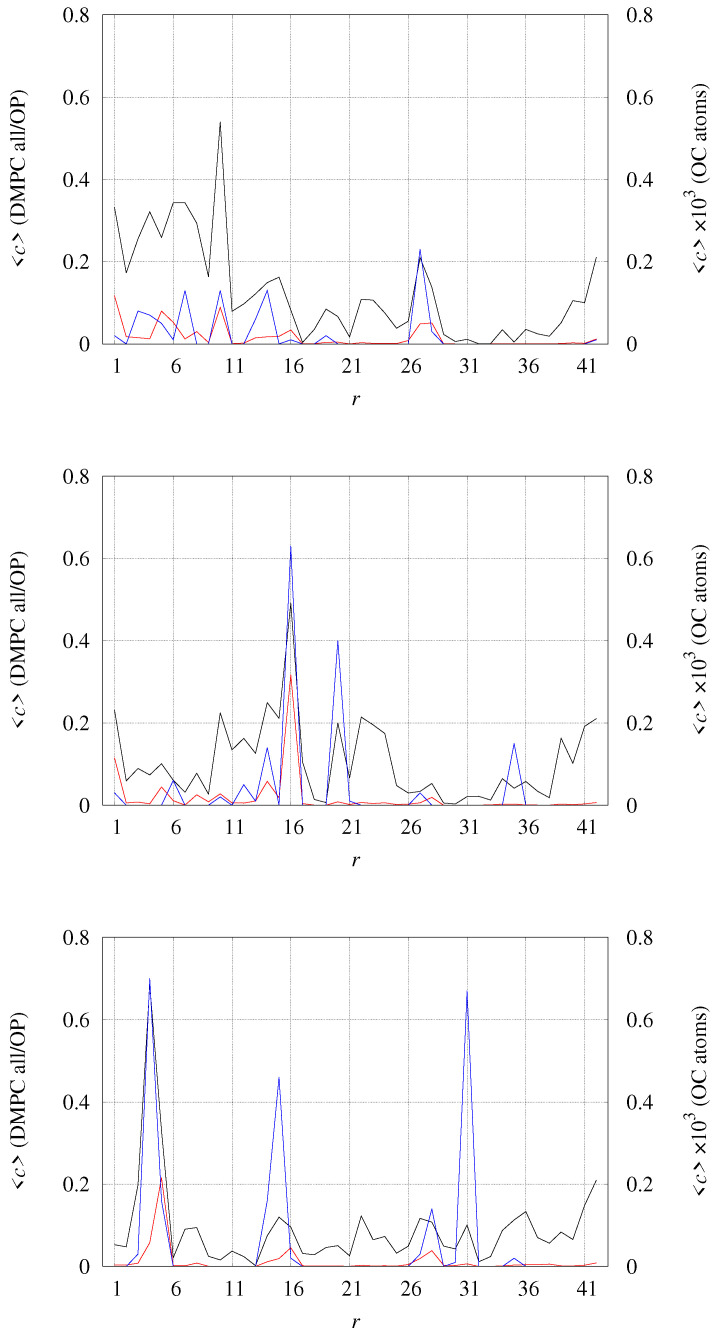
Average number of contacts *c* between different sets of atoms in DMPC molecules and any atom in each residue *r*. *c* is counted one when a single atom of residue *r* has distance d ≤ 4 Å from any of the atoms in the selected group of DMPC atoms. Black line (left *y*-axis)—any atom in DMPC; red line (left *y*-axis)—phosphoryl OP atoms in DMPC; blue line (right *y*-axis)—carbonyl OC atoms in DMPC. (**top**)—model **1** (2 × [2 × [A$\beta_{42}$]]; (**middle**)—model **2** (2 × [2 × [Cu-A$\beta_{42}$]]; (**bottom**)—model **3** (2 × [[Cu-A$\beta_{42}$]${}_{2}$].

**Figure 10 ijms-24-12698-f010:**
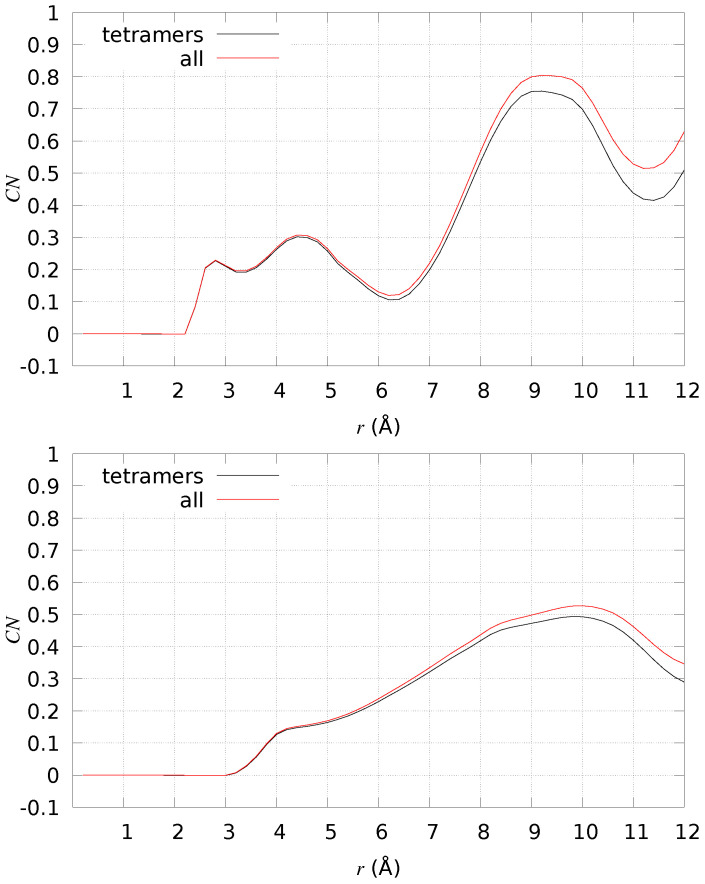
The difference in average number of particles (CN) at distance *r* for different sets of atomic pairs in A$\beta_{42}$. The difference is between 2 × [2 × [A$\beta_{42}$]] close to DMPC (+DMPC) and the same system in the water solution (−DMPC). Top panel—Nδ1/Nϵ2(His), all monomers, all possible pairs; bottom panel—Cδ(Glu 22)/Cγ(Asp 23)/C(Ala 42) on one side and Nζ(Lys 28)/N(Asp 1) on the other side, all monomers, all possible pairs. Black curve—counting all configurations of model **1** in water; red curve—counting only tetramers (R≤ 0.98) of model **1** in water.

**Table 1 ijms-24-12698-t001:** Summary of systems investigated in this study, with M indicating the addressed divalent cation in the list, Mg, Zn, Cu, Fe, and Ca.

System	Composition
Na/DMPC	1 Na^+^ + 9 K^+^ + 10 Cl^−^ + 64 DMPC + 3191 H_2_O
M/DMPC	1 M^2+^ + 8 K^+^ + 10 Cl^−^ + 64 DMPC + 3191 H_2_O
**1**	2 × [2 × A$\beta_{42}$] + 48 Na^+^ + 36 Cl^−^ + 25,013 H_2_PO
**2**	2 × [2 × Cu-A$\beta_{42}$] + 44 Na^+^ + 36 Cl^−^ + 25,017 H_2_O
**3**	2 × [Cu-A$\beta_{42}$]${}_{2}$ + 44 Na^+^ + 36 Cl^−^ + 25,017 H_2_O
**1**/DMPC	2 × [2 × A$\beta_{42}$] + 63 K^+^ + 51 Cl^−^ + 320 DMPC + 28,349 H_2_O
**2**/DMPC	2 × [2 × Cu-A$\beta_{42}$] + 59 K^+^ + 51 Cl^−^ + 320 DMPC + 28,353 H_2_O
**3**/DMPC	2 × [Cu-A$\beta_{42}$]${}_{2}$ + 59 K^+^ + 51 Cl^−^ + 320 DMPC + 28,353 H_2_O

**Table 2 ijms-24-12698-t002:** Summary of simulations performed in this study, with M indicating the addressed divalent cation in the list, Na, Mg, Zn, Cu, Fe, and Ca, and models as indicated in Table 1. Abbreviations: cMD is conventional molecular dynamics (MD); SMD is steered MD; US is umbrella sampling MD.

System	Type of	Number of	Time Length of
	Simulation	Trajectories	Each Trajectory
M/DMPC	cMD	1	1 μs
M/DMPC	SMD	1	1 μs
M/DMPC	US	1	0.7 μs
**1**	cMD	125	120 ns
**2**	cMD	128	120 ns
**3**	cMD	126	120 ns
**1**/DMPC	cMD	5	2.5 μs
**2**/DMPC	cMD	5	2.5 μs
**3**/DMPC	cMD	5	2.5 μs

**Table 3 ijms-24-12698-t003:** The 2–rank order parameters (pure numbers) between −½ and one) of Cα-H and Cβ-H bonds in DMPC in the presence of different ions M. The error in simulated averages (maximal root-mean square deviation among the probed lipid molecules) is of the order of 0.5. The experimental values are those reported for POPC [57] in the absence of divalent cations.

System	α	β	Exp. α	Exp. β
Na/DMPC	0.159730	−0.001753	0.05	−0.05
Mg/DMPC	0.158999	−0.002734	-	-
Ca/DMPC	0.154460	−0.003522	-	-
Fe/DMPC	0.159847	−0.002685	-	-
Zn/DMPC	0.158122	−0.003374	-	-
Cu/DMPC	0.165765	−0.002311	-	-

**Table 4 ijms-24-12698-t004:** Energy change in binding reaction (Equation 1).

M (Ion Type)	ΔE (kJ/mol)
Mg	67.7
Ca	−120.4
Zn	84.2
Cu	−32.3

## Data Availability

All data required to reproduce all of parts of the study are available upon request to authors.

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
