# Peer review of "Amyloid-β Tetramers and Divalent Cations at the Membrane/Water Interface: Simple Models Support a Functional Role"

_ijms, 2023, doi:10.3390/ijms241612698_

Round 1
Reviewer 1 Report
This manuscript describes results from molecular dynamics simulations of divalent cations next to a DMPC bilayer, and in a mixed system of an amyloid peptide tetramer next to a DMPC bilayer. The authors use a recent metal ion force field [ref 50] developed by a respected group. Also, they back up their findings by DFT calculations. Overall, the manuscript is a bit overloaded (to me) with results from different systems. It is hard to find a coherent story in this.
My main concern is whether the reported results are converged and robust.
(1) Line 343: “As observed above, all other cations can rarely, within the 1-µs MD simulation time, exchange the water molecules with atoms belonging to DMPC headgroups.” This sentence raised a warning flag in me. What is the average life time of contacts between divalent ions and particular oxygen atoms? Apparently, one cannot expect to observe many exchanges. What arguments can be given to convince the reader (and reviewer) whether the simulation results are converged?
(2) Section 3.4 starting in line 381: the analysis is very descriptive. It is unclear how reproducible and converged the findings of the authors are. With respect to table 1, did you run 128 or 5 replicas in each case? How consistent are the findings between replica simulations?
(3) Section 3.2: you do not show radial distribution functions “for clarity”. In contrast to this, I would prefer to see them, e.g. as supporting material.
Minor points:
Line 22: celll physiology -> cell physiology
Line 27: lower concentration than what?
Line 31: interactios -> interactions
Line 50: to to better -> to better
Line 113: neglectng -> neglecting
Line 131: “a little portion failed to achieve thermal conditions” is unclear. Does this mean the simulations crashed due to Shake/lincs errors?
Line 194: were the formation -> where the formation
Line 225: “Because of the simulation algorithm, peptides can not dissociate from the pre- 225 formed tetramers” : unclear to me, don’t you run unrestrained MD simulations where molecules are free to bind and unbind from each other?
Line 232: “the protein surface (P) can be hindered”: this should be reworded. Do you mean the surface is covered?
Line 288: thann -> than
Line 378: reword “, the initially OP Cu-bound atoms 378 become farther from Cu,”. Do you mean the distance increases?
Line 402: reword “that the Aβ tetramer interacts with the DMPC bilayer better”. More contacts is not “better” than fewer contacts. E.g. replace “better” by “more”.
Line 580: reword “was found more dramatic”. Reporting about MD results is not a drama.
Legend of Tab. 2 “Error in simulated averages (maximal root-mean square deviation among the probed lipid molecules) is of the order of 0.5.” is unclear to me because it has no unit. Do you mean 0.5 Angstroems? Or 0.5%?
Legend of Fig. 2: dispalyed -> displayed
Legend of Fig. 3 “the distance between the addressed cation M and the bilayer center calculated as the average distance between P atoms” is unclear. This is better defined in line 274.
I noticed a few glitches (see above). Overall, the quality of English is acceptable.
Author Response
Dear Reviewer 1,
Please, read the attached PDF file for a complete list of revisions.
See marks Q1.x/A1.x.

Reviewer 2 Report
Review report
Journal: IJMS (ISSN 1422-0067)
Manuscript ID: ijms-2510201
Type: Article
Title: Amyloid-β tetramers and divalent cations at the membrane/water interface: Simple models support a functional role
Authors: Pawel Krupa, Giovanni Penna*, Mai Suan Li
Section: Molecular Biophysics
Special Issue: Advanced Research in Functional Amyloids
General
In this manuscript, the authors focused on the study of charge polarisation at the membrane interface, a fundamental process in biology. In particular, they focused on the role of divalent cations (such as Ca2+, Mg2+, Zn2+, Fe2+, Cu2+) in specific compartments, such as the neuronal synapse. In this work, they applied a recent divalent cation model to a well-researched model of a polar lipid bilayer, DMPC. This model allowed a good description of the changes in the hydration of charged and polar groups as cations associate with lipid atoms. In addition, the authors modelled β-amyloid 1-42 (Aβ 42) peptides assembled in tetramers on the surface of the same bilayer. Two of the protein tetramer models were loaded with 4 Cu2+ ions, the latter bound as in the DMPC-free Aβ 42 oligomers. The two Cu-bound models differ in the binding topology: one with each Cu ion binding each of the monomers in the tetramer; one with pairs of Cu ions binding two monomers in dimers, forming tetramers as dimers of dimers.
In general:
1. The authors extended previous models suggesting a protective role of the Aβ-42 monomer at the synapse. First, they extended the description of interactions between a model polar lipid bilayer, DMPC, and divalent cations. They used the most recent cation model and for the first time applied this model to DMPC.
2. They found that free divalent cations perturb polar and charged bilayers mainly by strong charge neutralization. The bilayers themselves are more affected by amyloid fibrils than by amyloid peptides in soluble forms.
3. The authors also performed simulations of peptides, both in aqueous solution and near DMPC. These simulations provided a large number of possible collisions.
4. Overall, these results provide clues about the possible role of Cu ions in synaptic plasticity and of Aβ 42 oligomers in the removal of cations from the bilayer, with a particular focus on copper.
Minor points
The manuscript appears to be meticulously crafted, adhering to a logical narrative progression. The experimental design is robust, and the findings presented are compelling. I advocate for its publication, contingent upon the clarification of the following minor points:
1. The authors reference several studies that provide experimental evidence supporting the interaction of Cu+2 with DMPC (line 38). However, a study (that is not mentioned) that explored the Cu+2/Aß(1-42)/DMPC combination, yielding similar conclusions but through experimental means. I would advise the authors to elaborate on how the experimental evidence (in general) for this phenomenon aligns with the modelling approaches proposed in this manuscript.
References:
(i) Suwalsky et al., J. Inorg. Biochem. 1998, 70, 233-238. doi:10.1016/S0162-0134(98)10021-1. (already cited in the manuscript)
(ii) Suwalsky et al., Journal of Alzheimer's Disease, vol. 17, no. 1, pp. 81-90, 2009.
2. In the abstract, line 3, the authors refer to the "neural synapse" as a "compartment". I suggest revising this sentence to preclude potential misunderstandings.
3. On line 22, please replace "celll" with "cell".
4. In line 63, the authors assert: "The aim of this work is to understand the potential location of free divalent cations that may be released, for any given reason, in the synapse". If this is indeed the case: then why was the study of additional phospholipids not considered to render their system as analogous as possible to that of natural neurons?

Author Response
Dear Reviewer 2,
Please, read the attached PDF file with the complete list of revisions.
See marks Q2.x/A2.x.
